# Effect of Physical Activity on Obesity in Second Stage Pupils of Elementary Schools in Northwest Bohemia

**Jana Pyšná \*, Ladislav Pyšný, David Cihlář, Dominika Petrů and Martin Škopek**

Department of Physical Education and Sport, Pedagogical Faculty, Jan Evangelista Purkyně University in Ústí nad Labem, 400 96 Ústí nad Labem, Czech Republic; ladislav.pysny@ujep.cz (L.P.); david.cihlar@ujep.cz (D.C.); dominika.petru@ujep.cz (D.P.); martin.skopek@ujep.cz (M.Š.)

\* Correspondence: Jana.pysna@ujep.cz; Tel.: +420-606-545-623

**Abstract:** Obesity is a serious problem in our society. An evaluation of obesity development performed in the second half of the previous century already indicated a long-term positive trend in terms of body weight increase in children and the youth, which still persists today. Paediatric obesity arises from a changed lifestyle of children, characterised by an important restriction of their spontaneous physical activity. A lack of physical activity is one of the most important causes of paediatric obesity, which associated with a number of serious disorders. In the current study, the incidence of obesity and overweight as well as the relationship between physical activity and obesity in second stage pupils of elementary schools in northwest Bohemia is presented. The data collection was based on questions from the NAS 2001 questionnaire (nationwide anthropological survey) and BMI-for-age. 2001. NAS 2001 is a questionnaire for children and evaluates areas focused on engagement in physical activities and other daily activities, eating habits, drinking regime and care of the body habitus. Problems with obesity and overweight are present, particularly in boys. Only a third of boys and girls engage in sufficient physical activity. Differences were shown in the study group, where groups with higher BMI values had lower values of physical activity. Subsequently, a relationship was shown between those who use their bicycle as a means of transport and spend their leisure time bicycling at the same time. More than two-thirds of the study subjects reported using a bicycle as a means of transport and using their bicycle in their leisure time as a means of being active; 93% of these subjects had normal body weight. Our results confirm the continued pandemic prevalence of obesity and indicate that appropriate physical activity should be included in the everyday life of children both at school and outside of school.

**Keywords:** obesity; Body Mass Index; education; health; lifestyle; sport

## 1. Introduction

Due to evolution, the human body is prepared for relatively intensive physical stimulation that ensures its natural and healthy development. The necessary spontaneous activity is intensive and incorporates to an appropriate extent all physical abilities and skills of the child. Paediatric obesity arises from a changed lifestyle of children, characterised by an important restriction of their spontaneous physical activity. Obesity is a serious problem of our society where the rate of overweight and obese persons approaches 60% [1]. A risk situation can also be observed in children and adolescents. An evaluation of development performed in the second half of the previous century indicated a long-term positive trend in body weight gain in children and adolescents. Obesity has become an extensive problem, one aspect of which is the economic problem it poses for the entire healthcare system [2].

Obesity is a chronic, multifactorial disease. It develops due to a positive energy balance, particularly in persons with polygenic susceptibility to accumulation of fat. According to etiopathogenetic factors, it is most commonly divided into common obesity (which represents more than 90% of cases), monogenic obesity (obesity induced by certain drugs), obesity associated with endocrine diseases and hereditary syndromes [3–5]. With regard to the most prevalent form, common obesity, the deposition of fat is conditioned by many genes. According to Hainer et al. [3], more than 600 genes, markers and chromosomal regions are currently known to be associated with obesity. These genes affect, in particular, the regulation of food intake, including the feeling of hunger or satiety, eating behaviour with preference of certain foods, the absorption, processing, burning and deposition of consumed nutrients, but also the amount of hormones that regulate the energy balance [6].

The development of overweight and obesity is mostly affected by an increased percentage of fats in the diet given their high energy density, attractive taste, low satiation and postprandial thermogenesis [7]. More than 80 factors are involved in the development of obesity, including but not limited to intrauterine programming, prenatal and postnatal epigenetic factors, targeted selection of partners, some infections, short sleep duration or poor eating strategies [8–12].

Simple obesity is present in the vast majority of cases in our population, caused by an imbalance between energy intake and expenditure [3]. It is certainly affected by some genetic preconditions, as well; however, unfavourable epigenetic aspects are particularly crucial. If the child is influenced from early stages of its development (e.g. maternal obesity or smoking, course of the pregnancy, weight gain of the foetus, etc.), relatively unfavourable programming of the organism is the result. And if these changes are further supported by an unfavourably increased energy intake or inappropriate diet, they lead to excessive adipose tissue deposition at a young age. An ever decreasing level of spontaneous physical activity is thus found in children, together with a decline in their physical fitness and reduced motor development. On the contrary, obesity supports a temporary overall acceleration of the child's growth with acceleration of secular changes of the body [13–15].

Obesity and overweight represent a crucial current problem that poses a significant threat to the health of the population. Many serious diseases exist that are associated with, or may be a complication of overweight and obesity. These include, metabolic, cardiovascular, respiratory, oncological, gastrointestinal, orthopaedic and psychosocial complications as well as some endocrine disorders [16,17]. Unfavourable manifestations affecting the cardiovascular system pose the greatest threat to the lives of obese individuals, particularly those with abdominal obesity. Some physicians characterise these changes as the so-called global cardiovascular risk, where an increased amount and often also an inappropriate distribution of fat have an adverse impact on lipoprotein metabolism. Plasma concentrations of total cholesterol, triglycerides and low-density lipoprotein cholesterol (LDL-C) increase while the high-density lipoprotein cholesterol (HDL-C) level decreases. This developing dyslipidaemia supports atherosclerotic processes in the vascular system and increases cardiovascular risk several times—so-called atherogenic dyslipidaemia, e.g. Ipsen et al. [1], Vekic et al. [2] and Youseffi et al. [18]. Insulin resistance also plays a key role in the development of these changes. Its onset is probably driven by an interaction between genetic preconditions of the individual and some external factors. Besides some neurohumoral and immune mechanisms, the composition of the consumed diet is also important [19,20]. The above described changes result in the metabolic syndrome caused, in particular, by an increased adipose tissue mass, but also by changes in the muscle tissue or in the secretory portion of the pancreas [21,22]. Obesity, particularly visceral deposition of fat, also significantly increases the risk of type 2 diabetes mellitus [23,24]. A high risk of cardiovascular disease is associated predominantly with adverse atherosclerotic changes. These changes result from an inflammatory process with deposition of lipids in the vascular wall. It is triggered by a number of processes including inappropriate diet, abdominal obesity, dyslipidaemia and insulin resistance, but also by e.g. arterial hypertension or smoking [25–27]. These changes subsequently increase the risk of many serious clinical symptoms such as ischaemic heart disease, myocardial infarction or a cerebrovascular insult.

Obesity supports the development of other serious conditions, such as those affecting the motor and respiratory systems, or some cancer diseases. Increasing weight poses an enormous strain on the motor system. But even the adverse metabolic alterations caused by developing obesity affect the bone structure. This results in bone loss and adverse bone restructuring leading to decreased bone strength, e.g. Alberti et al. [21]. At the same time, increased weight poses a considerable mechanical strain on every joint of the obese individual, resulting in the development of osteoarthrosis [28]. Additionally, obesity has an adverse impact on the ventilatory function of the respiratory system due to, in particular, reduced elasticity of the thoracic wall and impaired contraction strength of respiratory muscles. Not only activity-induced dyspnoea, but also dyspnoea at rest is often found in obese individuals. This may, in some cases, even lead to obstructive sleep apnoea [22,29]. Furthermore, adipose tissue dysfunction may, most likely, have an adverse effect on the development of some tumours [30,31].

All these health risks increase in proportion to BMI values. An obese child is already affected by many health-related complications. Obesity in childhood, often persisting into adulthood, is a significant predisposing factor for the development of serious cardiometabolic and psychosocial complications and contributes to a shorter life expectancy [32].

Physical activity plays a crucial role in the prevention of obesity and diseases associated with the risk factors of obesity; it leads to the prevention of serious cardiovascular, metabolic and cancer diseases while at the same time improving the quality of life. If implemented to an appropriate extent, physical activity contributes to a reduction of weight and weight maintenance, and also to an improvement of metabolic complications [33,34]. Physical activity from an early age provides the most suitable way of primary prevention of obesity, i.e., at a time when a positive relationship and attitude to physical activity and to its lifelong practice are formed [35,36]. A considerable decrease in physical activity is currently observed in the life of an individual from childhood to adult age, particularly due to the sedentary lifestyle. However, decreasing physical activity has been demonstrated to be associated with an increase in obesity development [37]. Physical activity of moderate intensity for at least 60 min a day is recommended for children and adolescents from 7 to 15 years of age. Under these conditions, physical activity represents about 30–50% of daily energy expenditure and is one of the main factors contributing to a positive energy balance despite a reduced intake of energy and fats [38,39]. A recent study has found that only 2.5% of the students aged between 11 to 15 years old met the 60 min MVPA/day recommendation [40]. An adjustment of energy balance during growth is affected particularly by increasing energy expenditure through physical activity. According to Lavie et al. [34] Sattar et al. [41], the nature, sufficient intensity, frequency and duration are the decisive indicators. Dynamic, aerobic activities are most important for older children and adolescents, which also contribute to fat reduction in already obese individuals.

Appropriate goals in the management of excessive body weight are focused on achieving a realistic weight loss needed to reduce health-related risks. The goals also include an effort to maintain the achieved reduction and to prevent recurrent weight gain. The goals of obesity management are wider and include, in particular, an effort to reduce the risks and improve health. This can be achieved by weight reduction, optimally by 5–10% of the original body weight, particularly through increased physical activity and improved nutritional value of the diet. The role of physical activity in the reduction regimen depends on patient age, obesity degree and presence of health-related complications [18,42]. An essential goal of obesity management is to reduce metabolically active adipose tissue and to reduce the risks of health-related complications of paediatric obesity. This can be achieved even by a slight weight reduction and regimen measures. The treatment of children by physical activity should achieve, in particular, changes in their body composition; this means reduction of the body fat volume with concurrent increase in height, and development of active fat-free mass [43]. The prescribed physical activity must be individualized. The therapy is based on interdisciplinary cooperation of specialists [44–46].

Energy expenditure during physical activity depends on intensity and duration of the activity, on body weight and training, and also on neurohumoral and sympathoadrenal activation.

The adherence to recommendations leads not only to overweight and obesity reduction, but also to an improvement of the individual's cardiorespiratory and cardiovascular systems, muscle strengthening and improvement of bone health, as well as to a reduction of depression or anxiety [41,47].

The treatment with physical activity and increasing physical fitness and aerobic capacity provides the easiest method of utilization and reduction of fat reserves while developing muscle mass and vital organs [36,48]. Other authors, e.g. Gudzune et al. [49], Kasalová-Daňková et al. [50] and Yousefi et al. [18] report a number of positive changes after reduction therapy. These include, in particular, decreased BMI and body fat percentage, body circumference values, oxygen consumption and energy expenditure with the same comparable physical strain. These changes contribute to increased spontaneous physical activity and participation in physical educational programmes. At the same time, they are associated with a decrease in systolic and diastolic blood pressure and serum lipid concentrations. Furthermore, a decrease in leptin and insulin levels can be observed and the positive glucose tolerance also changes.

Physical activity impacts the energy metabolism, adipose tissue metabolism and metabolic complications associated with obesity. In the whole-body energy metabolism, physical activity changes the energy balance of the body by increasing energy expenditure; furthermore, it affects energy expenditure at rest and postprandial thermogenesis (energy needed to process and utilize consumed food), and the relative representation of fats when covering energy expenditure at rest as well as during physical activity. Overall energy expenditure increases with an increased amount of physical activity. The expenditure depends on the type and volume of physical activity while the key roles are played by its duration and intensity. And when regular physical activity is accompanied by an appropriate diet, its effect on energy expenditure at rest is even greater. A one-time physical strain in obese individuals also increases postprandial thermogenesis after the activity [51,52]. Physical activity has a clear positive impact on metabolic complications associated with obesity. This includes, in particular, insulin resistance, dyslipidaemia and hypertension. Regular physical activity reduces insulin resistance and this reduction persists for up to 72 h from the end of physical strain, and according to some studies, it becomes even more profound when accompanied by a reduction diet. Physical activity has a similar effect in individuals with hypertension where the post-exercise hypotension persists for up to 42 h [53].

The data that summarizes the development of obesity in 10-year intervals has been summarized by Prokopec [54] who showed a long-term positive trend in body weight increase. These measurements were built on by other authors following body characteristics of our children in 2001 [55,56]. As confirmed by their results, the percentage of obese boys increased in all age groups during the study period, while the excessive weight percentage increased only in schoolchildren. In the sample of girls, the percentage of both excessive weight and obesity increased in all studied age groups with the exception of the oldest age group. A relatively clear evaluation of the long-term trend (in 1951 to 2001) in overweight and obesity development in our 7-year-olds was also performed by Kunešová et al. [57,58]. As shown by the results, the percentage of overweight (including obesity) increased in both sexes; specifically, from 13.0% to 26.8% in boys and from 10.9% to 22.9% in girls. The evaluated prevalence of obesity increased from 1.7% to 8.3% in boys and from 1.7% to 6.9% in girls. Further development of overweight and obesity in our children was recently evaluated by the study "Zdraví dětí 2016" ("Children's Health 2016") in a sample of 5132 children in age groups of 5, 9, 13 and 17 years. The study sample included 8.1% low weight respondents, while 74.1% of respondents had normal weight, 7.5% were overweight and 10.3% were obese. Based on an evaluation of development during the past 20 years, the percentage of children with higher weight thus increased by 7% and the rate of obese children almost doubled. However, the rate of obese or overweight children remained the same between 2011 and 2016. The National Institute of Public Health (NIPH) [59] processed the development status in our children and adolescents for the past two decades. The results showed a considerably increasing higher than normal weight in our boys and girls. Summarizing the data, we can confirm that the values of overweight and obesity keep rising in our children [60].

As regards an evaluation of paediatric overweight and obesity prevalence in northwest Bohemia, we must note that no necessary data is available. Although some theses were found in available literature, which focused on the issue of obesity in this region, an intentional group of only several tens of individuals was described in all cases (e.g. Mikulencová [61], Sýkorová [62], Žitná, [63]. The thesis of Nosek [64] was the only partially more "extensive" study, which followed some anthropometric data in a group of 424 boys and 382 girls in 16 elementary schools and multi-year grammar schools in northwest Bohemia. The portion of overweight and obese children in this thesis represented 25.8% of the study individuals, while the rate of obese boys was 13.4% and that of obese girls was 8.1%.

On the contrary, data available for the global situation of overweight and obesity prevalence is extensive and the presented data is comparable in many cases, although sometimes also different from the situation in the Czech Republic. For example, the data of the World Health Organization related to the situation in Europe confirmed serious prevalence of paediatric overweight (including obesity) in 6-year to 9-year old children, ranging between 18% to 57% in boys and 18% to 50% in girls, while the prevalence of obesity reached 6% to 31% in boys and 5% to 21% in girls [65]. It is thus certain that we are facing a serious pan-European problem. Nevertheless, it should be kept in mind that large differences exist not only between individual evaluated European countries where obesity has been increasing, particularly in southern European countries, but also between specific regions. Obviously, paediatric obesity is also affected by a number of other external factors of the environment in which the child lives. The fact that the prevalence of obesity has been increasing due to these changes also in the Czech Republic, not only in children, but also in adults, is clearly an adverse observation [18,62].

The purpose of our paper is to enhance the knowledge of the current spread of overweight and obesity as assessed by the BMI, with respect to sports and physical activity in second-stage pupils of selected elementary schools in northwest Bohemia. Physical activity plays a key role in the prevention of obesity and diseases where it is involved as a risk factor.

## 2. Materials and Methods

The survey was performed in 1073 second stage pupils. The survey was carried out in 2019 and 2020. The schools were determined based on random selection from all elementary schools in northwest Bohemia. In the past, the northwest region of the Czech Republic was highly burdened by industrial activity, which still continues to exert a negative impact on the quality of the environment. The population density exceeds the national average, and the population of the region is characterized by a relatively young age of the inhabitants, the mean age being 41.6 years, and the highest mortality rate in the Czech Republic. The composition of the population is characterized by a high number of socially disadvantaged inhabitants. Out of the 34 selected schools, 17 agreed to the data collection. The survey was carried out in all second stage pupils of these schools who were present on the survey day. The pupils were examined during their physical education classes. Indicative data on the relationship to sports and physical activity was obtained from questions included in the NAS 2001 questionnaire [66]. The standardized NAS 2001 questionnaire for children and the youth evaluates areas focused on engagement in physical activities and other daily activities, eating habits, the drinking regime, and care of the body habitus.

The body weight was evaluated using calibrated scales and their physical height was measured using a calibrated gauge. Given that the Body Mass Index gradually increases from the end of preschool age until adult age, we used the "BMI-for-age" values [67] (Table 1). The determined data is presented in tables. The data collection was approved by the Ethics Committee of PF UJEP (Faculty of Education, J. E. Purkyně University) (1/2019/01) and was performed based on informed consent of the parents.

**Table 1.** Body weight classification according to BMI.

| Classification | BMI Value | Health Risk |
|---|---|---|
| Underweight | < 18.5 | Low |
| Normal weight | 18.5–24.9 | Average |
| Increased weight | ≥ 25 | Average |
| Overweight | 25–29.9 | Slightly increased |
| Obesity class I | 30.0–34.9 | Moderately increased |
| Obesity class II | 35.0–39.9 | Highly increased |
| Obesity class III | ≥ 40 | High |

The study group included 1073 pupils of elementary schools in northwest Bohemia aged 11–14 years, attending the 6th to the 9th classes, comprising 571 boys and 502 girls (Table 2). Mean age of the girls was 12.9 years and that of the boys was 13.1 years.

**Table 2.** Distribution of boys and girls in the group according to the school year.

|  | 6th Class | 7th Class | 8th Class | 9th Class |
|---|---|---|---|---|
| Boys, *n (%)* | 168 (29.42) | 144 (25.22) | 128 (22.42) | 131 (22.94) |
| Girls, *n (%)* | 146 (29.08) | 154 (30.68) | 113 (22.51) | 89 (11.73) |

The determined data is presented in tables and graphs. Significance tests were also used in addition to basic descriptive characteristics. The $\chi^2$ test for independence and the Anova test were used to evaluate the determined relationships and to determine any differences in BMI with respect to sex and physical activity. The null hypothesis was rejected with a probability of error below 5%, i.e., when the p value decreased below 0.05. The Shapiro Wilk test was used to determine normality of the data [68].

## 3. Results

It can be noted that in the studied group of second-stage pupils of elementary schools, problems with obesity and overweight were found predominantly in boys. We can confirm that these problems are more common in boys. Table 3 presents the distribution of boys and girls in the group according to their normal BMI.

**Table 3.** Distribution of boys and girls according to normal BMI.

|  | Under Weight | Normal Weight | Overweight | Obesity |
|---|---|---|---|---|
| Boys, *n (%)* | 28 (4.90) | 408 (71.45) | 124 (21.72) | 11 (1.93) |
| Girls, *n (%)* | 24 (4.78) | 423 (84.26) | 55 (10.96) | 0 (0.00) |

Pearson Chi-square: 33.88; df = 3; *p* = 0.0000001.

Table 4 presents mean SPA values per week (hours) in boys and girls. Based on this indicator, the boys and the girls were divided into sporting (seven and more hours of SPA in their leisure time) and not sporting (less than seven hours of SPA in their leisure time) as follows from Table 5.

**Table 4.** Mean SPA values per week (hours).

|  | Number | Mean | Median | Minimum | Maximum | Std. Dev. |
|---|---|---|---|---|---|---|
| Boys | 571 | 4.82 | 3.00 | 0.00 | 26.00 | 4.98 |
| Girls | 502 | 4.59 | 4.00 | 0.00 | 30.00 | 3.94 |

**Table 5.** Distribution of boys and girls according to SPA.

|  | **Sporting** | **Not Sporting** | **Total** |
|---|---|---|---|
| Boys, *n* (%) | 173 (30.30) | 398 (69.70) | 571 |
| Girls, *n* (%) | 156 (31.08) | 346 (68.92) | 502 |

Pearson Chi-square: 0.08; df = 1; $p$ = 0.7827241.

It can be noted that the percentage of boys and girls with physical activity is in the same order of magnitude; at the same time, none of the sexes was shown to have significantly more SPA hours in their leisure time ($p$ = 0.3884762).

Subsequently, we were interested in any possible relationship between SPA and normal BMI. The results are presented in Table 6.

**Table 6.** Relationship between SPA and normal BMI.

|  | **Sporting** | **Not Sporting** | **Total** |
|---|---|---|---|
| Underweight, *n (%)* | 18 (34.62) | 34 (65.38) | 52 |
| Normal weight, *n (%)* | 292 (35.14) | 539 (64.86) | 831 |
| Overweight, *n (%)* | 19 (10.61) | 160 (9.39) | 179 |
| Obesity, *n (%)* | 0 (0.00) | 11(100.00) | 11 |

Pearson Chi-square: 46.92; df = 3; $p$ = 0.0000001.

It can be noted that a relationship between SPA and BMI has been found, i.e., children with the recommended amount of physical activity, thus seven and more hours per week, have lower BMI values.

At the same time, we wanted to see whether any differences would be found between the duration of SPA in the groups of children in individual BMI categories. Tables 7 and 8 present basic SPA indicators for boys and girls and individual BMI categories. Table 9 provides the results of an Anova test—mean SPA duration (hours/week) with respect to normal BMI. Based on the determined data, it can be stated that differences have been demonstrated in the duration of SPA in individual BMI categories, which means that lower SPA values are found in groups with higher BMI.

**Table 7.** Mean SPA duration (hours/week) in boys with respect to their normal BMI.

|  | **Number** | **Mean** | **Minimum** | **Maximum** | **Std. Dev.** |
|---|---|---|---|---|---|
| Underweight | 28 | 5.28 | 0.00 | 21.00 | 5.59 |
| Normal w. | 408 | 5.18 | 0.00 | 26.00 | 5.32 |
| Overweight | 124 | 3.71 | 0.00 | 17.00 | 3.46 |
| Obesity | 11 | 3.27 | 0.00 | 6.00 | 2.28 |

**Table 8.** Mean SPA duration (hours/week) in girls with respect to their normal BMI.

|  | **Number** | **Mean** | **Minimum** | **Maximum** | **Std. Dev.** |
|---|---|---|---|---|---|
| Underweight | 24 | 5.08 | 0.00 | 14.00 | 4.21 |
| Normal w. | 423 | 4.57 | 0.00 | 30.00 | 3.92 |
| Overweight | 55 | 4.55 | 0.00 | 20.00 | 4.10 |
| Obesity | 0 |  |  |  |  |

**Table 9.** Anova test—mean SPA duration (hours/week) with respect to normal BMI.

|  | **SS Effect** | **MS Effect** | **SS Error** | **df Error** | **MS Error** | **F** | ***p*** |
|---|---|---|---|---|---|---|---|
| SPA amount | 153.77 | 51.26 | 21 812.96 | 1069.00 | 20.41 | 2.51 | 0.0472243 |

As follows from Table 10, a relationship has been demonstrated between individuals who use their bicycle as a means of transport and spend their leisure time bicycling at the same time, and normal BMI, i.e., these children have lower BMI values and 93% of them are found in the normal weight category.

**Table 10.** Relationship between individuals who prefer using a bicycle, and normal BMI.

| | Prefer Using Bicycle | Not Prefer Using Bicycle | Total |
|---|---|---|---|
| Underweight, *n* (%) | 51 (98.08) | 1 (1.92) | 52 |
| Normal weight, *n* (%) | 769 (92.54) | 62 (7.46) | 831 |
| Overweight, *n* (%) | 160 (89.39) | 19 (10.61) | 179 |
| Obesity, *n* (%) | 3 (27.27) | 8 (72.73) | 11 |

Pearson Chi-square: 64.17; df=3; *p* = 0.0000001.

## 4. Discussion

Based on evaluation of the trend in obesity and overweight incidence in our research survey in second-stage children of elementary school in northwest Bohemia, boys achieve higher body weight according to BMI categorization. Our results differ only slightly when evaluated, including normal values, together with the results of studies by Bláha et al. [55], Jíra [58], Kunešová et al. [57], Prokopec [54], Vignerová et al. [56]. Optimal weight was found in 71% boys and 84% girls. On the contrary, higher than optimal weight was found in 22% boys and 11% girls.

The research survey of the group of 1073 pupils, 571 boys and 502 girls, of northwest Bohemia was undertaken to determine the incidence of obesity and overweight as well as the relationship between physical activity on one hand and obesity and overweight on the other. Our evaluation of obesity and of the relationship between obesity and physical activity provided the following results. Problems with obesity and overweight are present predominantly in boys (23%) compared to girls (11%). Both boys and girls of the study group were engaged in about the same percentage of physical activity. Only 30% girls and 31% boys reported being engaged in physical activity (more than seven hours a week excluding physical education classes). Comparing the amount of physical activity and BMI values, it can be noted that we have found a relationship between physical activity and BMI, i.e., differences in terms of physical activity duration were demonstrated in individual BMI groups – groups with higher BMI values showed lower values of physical activity. Subsequently, we demonstrated a relationship between individuals who use their bicycle as a means of transport and spend their leisure time bicycling at the same time. More than two thirds of the subjects reported using their bicycle as a means of transport and using their bicycle in their leisure time as a means of being active; 93% of these subjects had normal body weight.

At the same time, we demonstrated the expected favourable relationship between physical activity and observed BMI values. Higher BMI values were found in pupils not engaged in the required weekly physical activity. Clearly, no markedly different, higher-risk values were found in the second-stage pupils of the selected elementary schools in northwest Bohemia compared to the data of the "Children's Health 2016" [59]. For example, these findings have been demonstrated by 48 Hainer [3], Lobstein et al. [69] and Stranavska et al. [70] who emphasized regular physical activity as a major factor of health and of a healthy lifestyle for maintaining the functions of the body and also as one of the obesity prevention factors. It is also important to realize that physical activity, particularly on the level of fitness training, is now becoming a compensation of insufficient physical strain and mental pressure as a consequence of the contemporary predominantly sedentary way of life.

Our results also indicate that only one third of the pupils were physically active, while the percentage was about the same in boys and girls—30% for boys and 31% for girls. This information provides evidence of preferred leisure time spending e.g. by engaging in computer technologies or watching TV. This claim is based on the results of the research project Jíra [58] where the young people reported watching TV as their most favourite leisure time activity. Also, we can confirm the results

of the STEM/MARK [71] research project; this project provided interesting results—namely that on average, every week children spend 11 h watching TV, 5 h and 20 min on the computer, 5 h performing non-organized activities and only 2 h in hobby groups.

Furthermore, our results demonstrated a relationship between the amount of physical activity and BMI. The higher the physical activity rate, the lower the BMI values in the study group. These results provide evidence that besides the so-called family therapy associated with diet modification, it is increased physical activity that is one of the most important effects in childhood obesity prevention and management. The results indicating a favourable impact of physical activity on an individual's weight were further supported by yet another finding, namely the relationship between bicycling and BMI values. More than two-thirds of the subjects reported using their bicycle as a means of transport and also in their leisure time as a means of spending their leisure time actively and with their family; 93% of these subjects had normal weight. This data is also supported by the findings of Forýtková [72]; according to the results of scientific exploration of these authors, the role of parents in the context of paediatric obesity is robust, global and multidisciplinary. Besides dietary habits, all possibilities that will support predominantly movement stimulation in any potential "extra-curricular" activities of children should be activated. Physical activity habits learned during childhood are carried over to adulthood. As confirmed by extensive research of Telama et al. [73], regular physical activity performed between 9–18 years of age considerably increases the probability that the individual will also be active in adulthood. The probability of an active lifestyle in adulthood increases proportionally to the time spent doing sports in childhood. We also agree Whitehead [74] who reported that the support and motivation of parents and pedagogues was essential for adequate physical activity of the children. It is specifically intrinsic motivation that has been found to be related to the daily moderate-to-vigorous physical activity [40].

Physical skills or physical literacy should be included in the five essential types of literacy. According to its concept, physical literacy should lead to a desire to be active, improve one's own physical skills, develop abilities and try new forms of physical activity. In this respect, Martinek [75,76] reported that physical activity showed a strong relationship with body weight reduction but that obese individuals often lacked motivation in their effort to reduce their body weight. This is precisely why the role of parents and pedagogues is so significantly motivational.

In our research survey we used BMI to evaluate overweight and obesity; this approach is useful for examination of a larger group of respondents as regards its quickness. BMI is a globally recognized index applicable to all age categories [7,42]. Indices calculated from anthropometric parameters, such as BMI, should be evaluated based on national percentile graphs and the dynamics of development of their values should be followed in time. We are aware that providing an inaccurate result for persons with a non-standard portion of any body tissue, e.g. those with an above-average muscle tissue percentage, is a drawback of BMI. Such individuals were not included in our research survey. More accurate but also more time demanding methods would be more appropriate for a more accurate evaluation of body composition and of obesity, including but not limited to calliper measurement or the bioelectrical impedance analysis method [72].

## 5. Conclusions

Besides the role of the family, the role of the school is also important in obesity management in children; the school should create an active environment for the prevention of overweight and obesity in children. This can be achieved by establishing a connection between education and a healthy lifestyle, specifically nutrition and physical activity on one hand and practical realisation on the other, and physical activity of the children also out of regular school classes [77]. Realistic goals with respect to physical activity and dietary habits are a precondition of successful prevention and management of obesity. The purpose is to develop an interest in physical activity through new sports and exercises. To ensure its sustainability, it is necessary to create a need of such a regular physical activity that the children will like, that will be easy to realise, while supporting desirable energy

expenditure. The development of proper dietary and physical activity habits can be facilitated through training in cognitive-behavioural techniques. For example, to train the child and their family not to pay excessive attention to eating, weight and body shape. The need of physical activity in childhood is physiologically higher than in adult age and it is the most appropriate approach in the prevention and management of obesity. It is thus necessary to eliminate the results of lifestyle changes in our children that lead to further metabolic changes in the body, associated with, in particular, increasing overweight and obesity, both in the family setting and at school [78].

When physical activity is indicated, it is necessary to observe the principles for a positive health effect of physical activity. The essential FITT rule—meaning frequency, intensity, time and type of activity—should be observed. Aerobic exercise is crucial in weight reduction. This type of activity provides sufficient oxygenation of the body, and after a certain period of time during which the glycogen reserves are used up, adipose tissue starts being processed as a preferred energy substrate. At the same time, the ketone levels start rising; ketones are a product of fatty acid degradation, which reduce the feeling of hunger. The intensity can be objectively determined by measuring the heart rate. If the main goal of the patient is to reduce adipose tissue, moderate intensity of physical activity is preferred, i.e., 40–65% of VO2 max. [34,51]. In the optimal range suitable for weight reduction, the body utilizes fat reserves as a source of energy; the intensity of exercise, expressed as heart rate, should correspond to 50–75% of the maximum heart rate (MHR). The indication of physical activity should be individualized; generally, higher values are more appropriate for younger, trained and especially healthy individuals, while lower values should be used for older persons, convalescents or beginners [47,73].

According to the guidelines of the European College of Sports medicine, US National Institutes of Health and the American College of Sports Medicine, the physical activity volume with an energy expenditure of 4200 kJ/week is sufficient to provide a beneficial health effect [79,80]. Physical activity of moderate intensity, with a duration of 150–250 min per week, leads to a mild weight loss. More profound changes in terms of weight loss can be achieved using physical activity taking longer than 250 min per week. The time of 250–300 min per week is sufficient to prevent weight gain, or 300 min per week combined with low-energy diet after previous weight reduction [81,82]. According to available research results, it is currently recommended to combine aerobic exercise with dynamic force training and force activity [43,83]. An optimal combination of aerobic and force training has been shown to be the most appropriate method leading to weight reduction and to an overall improvement or maintenance of a good physical condition [84–87].

**Author Contributions:** Conceptualization, J.P., L.P.; D.C.; methodology and study design, J.P., D.C.; data collection, J.P., D.P., M.Š.; data analysis, D.C.; data Interpretation, J.P. and L.P.; writing—original draft preparation, J.P. and L.P.; writing—review and editing, J.P., D.C., D.P.; visualization, J.P. and D.C.; project administration, J.P.; supervision, L.P. All authors have read and agreed to the published version of the manuscript.

**Funding:** This research was funded by [Lifestyle and the effect of the educational process on the health in Second Stage Pupils of Elementary Schools in the region Ústí nad Labem.], grant number [UJEP-IGA-TC-2019-43-01-2].

**Conflicts of Interest:** The authors declare no conflict of interest.

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
