# Peer review of "Effect of Physical Activity on Obesity in Second Stage Pupils of Elementary Schools in Northwest Bohemia"

_sustainability, doi:10.3390/su122310042_

Round 1

Reviewer 1 Report

No clear that is so special in the region of Ústí nad Labem to make include it in the paper title.
Do you expect other results in other cities?

The paper is a collection of data and results are no far from common sense. In other words, I do not understand how the paper can be considered a scientific paper and what are the scientific outcomes. No useful or surprising correlation. It worth to be mention that guys are usually more fat than girls, and who exercise more is less overweight.
Why more boys than girls in the analysis?
Why not an exact number of sportive guys both for the girls and the boys? This would simplify the analysis.

I believe that some charts to express the tables' data could make the data more clear at a first sight. Please add the respective chart of the tables

Please make explicit the weight type (from underweight to obese) with the respective BMI range.

Why the specif mention of the use of bicycle or not?

Discussion and conclusion section is plenty of general considerations more than focusing on the results of their research.

Author Response

Dear reviewer, 

i would like to thank the reviewers for their comments.a 

I send the "Response to Reviewer 1 Comments".

Best Regards,

Jana Pyšná

Reviewer 2 Report

Comments to the Authors

This is fairly well written paper. I have read this paper with much interest. Also, please see some concerns related to this paper.

Title

The title contains most of the key features of the article.

Abstract

Abstract is mostly well written, please see some minor comments.

Specific comments:

“A long-term positive trend in body weight gain continues to persist.” – please specify among whom?

“In their research survey, the authors determine the…” – please change that style. For example, use “In the current study, the incidence of obesity …”

“…in second stage pupils of elementary schools in the region of Ústí nad Labem.” – repetition.

“CAV 2001” – not sure what that is. Please (shortly) introduce that to the reader.

“As revealed by the research survey” – survey is redundant.

Introduction

The authors provide adequate review of the existing literature. Most of the recent and relevant studies are included. Overall, the introduction is clear and concise.

Specific comments:

“Obesity is a serious problem of our society where the rate of overweight and obese persons approaches 60%.” – I believe this needs to be cited.

“Simple obesity is present in the vast majority of cases in our population, caused by an imbalance between energy intake and expenditure.” – this also needs to be cited.

“the so-called atherogenic dyslipidaemia, e.g. 1 Ipsen et al. …” – is that ‘1’ a typo?

“Physical activity of moderate intensity for at least 60 minutes a day is recommended for children and adolescents from 7 to 15 years of age.” – I would highlight here that a recent study has found that only 2.5% of the students aged between 11 to 15 years old met the 60 min MVPA/day recommendation (Kalajas-Tilga et al., 2020).

Kalajas-Tilga, H., Koka, A., Hein, V., Tilga, H. & Raudsepp, L. (2020). Motivational processes in physical education and objectively measured physical activity among adolescents. Journal of Sport and Health Science, 9(5), 462–471. https://doi.org/10.1016/j.jshs.2019.06.001  

Materials and Methods

Specific comments:

“questions of the CAV 2001 questionnaire” – please introduce more specifically this questionnaire.

“The X2 test for independence and the Anova test were used to evaluate the determined relationships and to determine any differences.” – please specify which differences are expected?

Results

Specific comments:

Please check the formatting of all the tables in accordance with the journal guidelines.

Please be consistent on how many decimal places are presented throughout the manuscript.

Table 6 and Table 10 – I believe p value cannot be equal to the zero.

Table 9 – not sure if a table is necessary if there is only one line. Also, table 9 is not referred in text.

Discussion

The authors have discussed the results from multiple angles and placed it into proper context without being overinterpreted. The authors link their findings to previous studies.

Specific comments:

“report_ed” – please check this, first line on page 10.

Specific comments: “We also agree Whitehead [73] who reported that the support and motivation of parents and pedagogues was essential for adequate physical activity of the children.” – I believe it should be acknowledged that it is specifically intrinsic motivation that has been found to be related to the daily moderate-to-vigorous physical activity (Kalajas-Tilga et al., 2020).

Kalajas-Tilga, H., Koka, A., Hein, V., Tilga, H. & Raudsepp, L. (2020). Motivational processes in physical education and objectively measured physical activity among adolescents. Journal of Sport and Health Science, 9(5), 462–471. https://doi.org/10.1016/j.jshs.2019.06.001  

Author Response

Dear reviewer, 

i would like to thank the reviewers for their comments.a 

I send the "Response to Reviewer 2 Comments".

Best Regards,

Jana Pyšná

Round 2

Reviewer 1 Report

The paper has been improved and it is now valuable for submission in Sustainability - MDPI